

# Explicating anti-amyloidogenic role of curcumin and piperine via amyloid beta (Aβ) explicit pathway: recovery and reversal paradigm effects

Aimi Syamima Abdul Manap[1], Priya Madhavan[2], Shantini Vijayabalan[3], Adeline Chia[1] and Koji Fukui[4]

[1] School of Biosciences, Faculty of Health and Medical Sciences, Taylor's University, Subang Jaya, Selangor, Malaysia
[2] School of Medicine, Faculty of Health and Medical Sciences, Taylor's University, Subang Jaya, Selangor, Malaysia
[3] School of Pharmacy, Faculty of Health and Medical Sciences, Taylor's University, Subang Jaya, Selangor, Malaysia
[4] Department of Bioscience and Engineering, College of Systems Engineering and Science, Shibaura Institute of Technology, Saitama, Japan

Corresponding authors
Priya Madhavan,
Priya.Madhavan@taylors.edu.my
Koji Fukui, koji@shibaura-it.ac.jp

## ABSTRACT

Previously, we reported the synergistic effects of curcumin and piperine in cell cultures as potential anti-cholinesterase and anti-amyloidogenic agents. Due to limited findings on the enrolment of these compounds on epigenetic events in AD, we aimed at elucidating the expression profiles of Aβ42-induced SH-SY5Y cells using microarray profiling. In this study, an optimized concentration of 35 μM of curcumin and piperine in combination was used to treat Aβ42 fibril and high-throughput microarray profiling was performed on the extracted RNA. This was then compared to curcumin and piperine used singularly at 49.11 μM and 25 μM, respectively. Our results demonstrated that in the curcumin treated group, from the top 10 upregulated and top 10 downregulated significantly differentially expressed genes ($p < 0.05$; fold change $\geq 2$ or $\leq -2$), there were five upregulated and three downregulated genes involved in the amyloidogenic pathway. While from top 10 upregulated and top 10 downregulated significantly differentially expressed genes ($p < 0.05$; fold change $\geq 2$ or $\leq -2$) in the piperine treated group, there were four upregulated and three downregulated genes involved in the same pathway, whereas there were five upregulated and two downregulated genes involved ($p < 0.05$; fold change $\geq 2$ or $\leq -2$) in the curcumin-piperine combined group. Four genes namely *GABARAPL1*, *CTSB*, *RAB5* and *AK5* were expressed significantly in all groups. Other genes such as *ITPR1, GSK3B, PPP3CC, ERN1, APH1A, CYCS* and *CALM2* were novel putative genes that are involved in the pathogenesis of AD. We revealed that curcumin and piperine have displayed their actions against Aβ via the modulation of various mechanistic pathways. Alterations in expression profiles of genes in the neuronal cell model may explain Aβ pathology post-treatment and provide new insights for remedial approaches of a combined treatment using curcumin and piperine.

## INTRODUCTION

In the pathogenic cascade of Alzheimer's disease (AD), misfolding, aggregation and deposition of amyloid $\beta$ (A$\beta$) peptides in the brain parenchyma and vessel walls lead to severe consequences (*Watson et al., 2005*). Over the past couple of decades, several studies have highlighted that the A$\beta$ aggregates as being the core determinants in molecular mechanisms contributing to AD (*De Felice et al., 2008*; *Guglielmotto et al., 2014*). In addition, it was proposed that there are different A$\beta$ assemblies, each characterized by different molecular sizes, stability and neurotoxic characteristics (*Jin et al., 2011*). However, their particular significance to AD pathogenesis is uncertain. Natural products, in which their phytochemicals are known to have numerous beneficial biological neuroprotective effects, are of specific concern to scientists in this era (*Bui & Nguyen, 2017*).

We previously demonstrated neuroprotective effects of combined treatment with curcumin and piperine against A$\beta$ induced degeneration by in silico and in vitro assays (*Manap et al., 2019*). Curcumin and piperine at 35 μM in combination were able to inhibit neurotoxicities, aggregation and disaggregate A$\beta$ fibrils as well as reversed A$\beta$-induced neuronal oxidative stress (*Manap et al., 2019*). In the present study, we continue our investigation at a molecular level by using high-throughput microarray technology in order to elucidate differences in the gene expression profiles between AD and treatment groups. Gene expression microarray offers a new tool to address complexity, allowing for overviews of concurrently multiple cellular pathways. The main benefit of the microarray approach is the capacity to explore thousands of genes of interests simultaneously, although low statistical power, elevated false positives or false negatives and unclear reference to functional endpoints often hinder data interpretation.

A large number of expression profiles was examined in the compound-specific group. We observed that genes that were altered and are involved in the mechanism of A$\beta$ appeared in different treatment groups, which signifies that both single and combined compounds exerted neuroprotective activities against the degeneration of A$\beta$. Nevertheless, we found common genes that were differentially expressed in all single and combined treatment with curcumin and piperine. The genes *GABARAPL1*, *CTSB*, *RAB5* and *AK5* had shown to be involved in the intrinsic pathway that modulates the processing of A$\beta$ including macroautophagy and neurite degeneration. In addition, in regards to treatment with a single compound, when focusing on the A$\beta$ pathway in AD, we revealed an explicit pathway that modulates the expression of A$\beta$ level and interferes with AD progression. These genes were *PICALM, LRP1, CTSB, ADAMTS5*, *APOE* and *PSEN1,* which showed to be involved in endothelial A$\beta$ trafficking and disruption in A$\beta$ production or rapid A$\beta$ clearance. Next, we further sought whether the single and combined treatment with curcumin and piperine were able to exert reversal actions on the damages caused by A$\beta$. Interestingly, regardless of whether it was single or combined treatment, we found protective genes which restored synaptic losses caused by A$\beta$ via synaptic modulation, restoration of ubiquitin proteasome system (UPS) pathway, a reversal of neuronal apoptosis and neurite degeneration. We also demonstrated from our study that novel putative genes with limited literature in AD pathologies such as *TGIF1, IGFBP3, LBH, ITGA9, SPRY1, VIM, INA, LYN, PLCB4*

and *OLFM1*. These genes appeared in our top 10 list of upregulated and downregulated genes in single and combined treatment with curcumin and piperine. The results of protein-protein interaction also demonstrated the significant pathways that were involved or potentially involved in the AD pathway, long-term potentiation, TGF-beta signaling pathway, dopaminergic synapse and others. These findings support the hypothesis that single and combined treatment with curcumin and piperine in SH-SY5Y cells exposed to A$\beta$ fibrils may result in differences in expression profiles of genes. Accordingly, a detailed analysis of how these compounds exerted their mechanism of action at the molecular level, the involvement of various potential genes and pathways in AD may lead to a novel finding in AD pathology. The data discussed in this paper have been deposited in NCBI's Gene Expression Omnibus (GEO) and are accessible through GEO series accession number: GSE143998.

## MATERIALS & METHODS

### A$\beta$ fibril preparation

Synthetic A$\beta$42 peptide was purchased from American Peptide (Sigma, USA) and prepared following the protocols described previously (*Caballero et al., 2016*; *Tycko, 2018*) with some modifications (Fig. 1). In brief, the A$\beta$42 peptide was dissolved to 1 mM in 100% 1, 1, 1, 3, 3, 3-hexafluoro-2-propanol (HFIP, Sigma) and aliquoted in non-siliconized polypropylene vials. The tubes were left in the fume hood overnight to remove HFIP. The traces of HFIP were removed under vacuum (Speed Vac) (Thermo Fisher Scientific, US) on the following day and re-suspended in dimethyl sulfoxide (DMSO) to a concentration of 5 mM. To form fibrillar conditions, the peptide was diluted to a final concentration of 100 $\mu$M with 10 mM of acid hydrochloric (HCl) solution and incubated at 4 °C for 24 h.

### Thioflavin T microscopy staining

In order to confirm the uptake of A$\beta$ in the cell, Thioflavin T (ThT) fluorescence assay was performed as described previously (*Jin et al., 2016*). Thioflavin T is a benzothiazole dye that shows increased fluorescence when binding to amyloid fibrils and is frequently used for the detection of amyloid fibrils (*Sulatskaya et al., 2018*). Initially, ThT was dissolved in 50% ethanol to 1 mg/mL and stored at 4 °C. For live cell imaging, cells treated with A$\beta$ fibrils were incubated with ThT at 10 $\mu$g /mL in DMEM (ATCC® 30-2002™) cell culture media for 30 min at 37 °C and examined via live cell fluorescence imaging. The cellular accumulation of ThT was assessed using Nikon's NIS-Elements fluorescence microscope (Nikon, Tokyo, Japan) and images were processed and analyzed with ImageJ digital image processing (USA).

### Immunofluorescence assay

Immunofluorescence (IFC) assay was performed in order to double confirm the presence and deposition of A$\beta$ fibrils in the cells. SH-SY5Y cells were seeded at a density of $5 \times 10^4$ cells/ml in $\mu$-slide 8-well IbiTreat chamber slides (Ibidi GmbH, Martinsried, Germany) and incubated at 37 °C in a humidified 5% $CO_2$ incubator. After the cells reached 80% confluency, the cells were treated with 25 $\mu$M A$\beta$ fibrils for 24 hrs. Cells were then washed

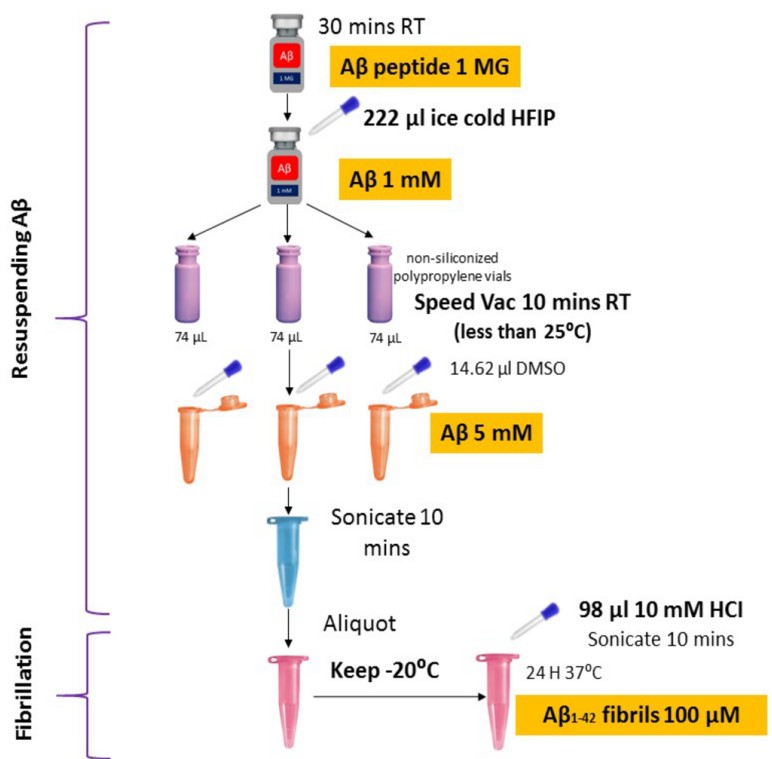

**Figure 1** Optimized amyloid-beta fibril preparation (Adapted with slight modifications from *Caballero et al., 2016*).

with PBS three times, fixed with 4% paraformaldehyde (Aldrich, Steinheim, Germany) for 15 min, and rewashed with PBS three times. Fixed cells were permeabilized with 0.1% Triton X-100 (Sigma-Aldrich, US), diluted in PBS for 15 min on ice, and washed three times with PBS. Non-specific binding was blocked with 10% bovine serum albumin (BSA; normal goat serum, AbCam, Cambridge, UK) diluted in PBS+ 0.1% Tween 20 for 1 hr at room temperature. Cells were incubated with mouse monoclonal [DE2B4] to beta amyloid (AB11132, 1:200 dilution, AbCam, Cambridge, UK) diluted in 1% blocking buffer (PBS + 0.1% Tween 20–1% BSA) overnight at 4 °C. Cells were then washed with PBS + 0.1% Tween 20 three times and incubated with secondary antibodies goat anti-mouse IgG H&L (Alexa Fluor® 488) preadsorbed (AB150117, 1:1000 dilution, AbCam, Cambridge, UK) diluted in 1% blocking buffer (PBS + 0.1% Tween 20–1% BSA) for 1 hr at room temperature. After washing three times with PBS + 0.1% Tween 20 in the dark, cells were incubated with fluoroshield mounting medium with DAPI (AB104139, AbCam, Cambridge, UK) for 5 mins at room temperature in the dark. Finally, cells were ready to be viewed under Nikon's NIS-Elements fluorescence microscope (Nikon, Tokyo, Japan). The images were processed and analyzed with ImageJ digital image processing (USA).

## Compound preparation

The preparation of the optimal concentration of pure curcumin, piperine and their mixtures was performed based on an optimized protocol (*Manap et al., 2019*).

## Cell culture and treatment

Human Neuroblastoma cells (SH-SY5Y) were purchased from ATCC, USA. The cell lines were maintained in DMEM (ATCC® 30-2002™), supplemented with 10% FBS, 5% penicillin/streptomycin and incubated at 37 °C in a humidified 5% $CO_2$ incubator. Cells at 80% confluency were seeded into 6-well plates at a density of $1.2 \times 10^6$ cells/ml. Cells were allowed to adhere overnight at 37 °C with 5% $CO_2$. On the following day, cells were treated with individual and combined compounds and incubated for a further 24 h. There were 4 groups in this study, i.e., one which had only A$\beta$ (as a control at 25 μM), second group had cells treated for 24 h with curcumin (49.11 μM) followed by addition of A$\beta$ (25 μM) and incubated for another 24 hrs, third group had cells treated for 24 hrs with piperine (25 μM) followed by addition of A$\beta$ (25 μM) and incubated for another 24 hrs and fourth group had a mixture of curcumin and piperine (35 μM), treated for 24 hrs followed by addition of A$\beta$ (25 μM) and incubated for another 24 hrs. These experiments were performed in triplicates.

## Total RNA extraction

Total RNA extraction was performed by using RNApure High-purity Total RNA Rapid Extraction Kit (Bioteke, China) according to the manufacturer's protocol. Total RNA concentration and purity were determined and samples were stored at −80 °C. All extracted RNA samples were subjected to spectrophotometric measurement (NanoDrop Spectrophotometer ND2000C, Thermo Scientific) and the RNA quality was determined using an Agilent 2100 Bioanalyzer according to the manufacturer's protocol.

## Microarray profiling

A total of twelve RNA samples were processed according to the Applied Biosystems™ recommended protocol. Briefly, 100 ng of total RNA was reverse transcribed to produce cDNA/mRNA hybrid molecule, which was subsequently used as a template to create double-stranded cDNA. This double-stranded cDNA was then amplified via in vitro transcription (IVT) to produce cRNA. In vitro transcription (IVT) generated cRNA was then purified and subjected to 2nd-cycle single-stranded sense cDNA synthesis which was later fragmented, labeled, and hybridized to Human Clariom S Array for 16 hrs at 45 °C with rotation at 60 rpm. Arrays were then washed and stained using the FS450_0007 fluidics protocol and scanned using an Applied Biosystems™ GeneChip™ Scanner 3000 7G.

## Array and data QC

The scanned images were inspected for hybridization efficiency and CEL files generated from GeneChip Command Console Software were imported into Transcriptome Analysis Console v4.0 software for array QC. RMA normalization was performed on the samples to generate the quality control (QC) metrics that was used to determine data quality. These include, all Probeset mean, background mean (Bgrd_Mean), positive and negative

probes (POS vs NEG AUC), bacterial spike controls and Poly-A controls. Protein-protein interaction network analysis visualization and pathway analysis were conducted by using NetworkAnalyst 3.0 (*Zhou et al., 2019*) and WikiPathways.

### Validation of genes by Real-time PCR (qPCR)

Ten genes were chosen from microarray analysis to be validated by qPCR namely, *SYPL1, RAB5, AK5, PICALM, CAP8AP2, APOE, GABARAPL1, PSEN1, CREB1* and *ADAMTS5*. The reverse transcription kit was used to synthesize the first-strand cDNA (ReverTra Ace qPCR RT Master Mix with gDNA Remover (Code No. FSQ-301). Real-time PCR was carried out using PrimeTime® Gene Expression Master Mix (IDT, USA) at 1X concentration containing PrimeTime® qPCR primers and 3 pg to 100 ng cDNA template. PrimeTime Standard qPCR Assay (5′–3′Dye-Quencher Mod: 6-FAM/ZEN/IBFQ) primers (*GAPDH* and *ACTB*) were used as an endogenous control to quantify the target genes. The final volume of each RT-qPCR reaction was 20 µL, which contained 10 µL PrimeTime® Gene Expression Master Mix (IDT, USA), 1 µL of each PrimeTime® qPCR Assay primer (IDT, USA), 2 µL of diluted cDNA template and 7 µL of nuclease free water. PCR cycling protocol was performed by using Eppendorf Mastercycler Realplex2 (Eppendorf, Germany). Cycling included polymerase activation step of 3 min at 95 °C was followed by 45 cycles of 5 s at 95 °C and 30 s at 60 °C. Data analysis on expression levels were calculated using the $2^{-\Delta\Delta Ct}$ comparative CT method (*Schmittgen & Livak, 2008*). The means and standard deviations were calculated from experiments performed in triplicate and are presented as the n-fold differences in expression.

## RESULTS

### Aβ42 fibrils detection using Thioflavin T staining

We had previously reported on the Aβ inhibition and disaggregation assay of selected compounds by using Thioflavin T fluorescence assay (*Manap et al., 2019*). Thioflavin T (ThT) is a small molecule that emits strong fluorescence upon binding to amyloids (*Xue et al., 2017*). Here, we showed that ThT also works as a dye which stains the Aβ42 fibrils (green fluorescence) in the neuronal cells treated with Aβ42 fibrils for 24 h (Fig. 2B). While for untreated cells (without the Aβ42), we could not see any fluorescence dye being emitted, which confirmed the absence of the fibrils (Fig. 2A). We demonstrated that the prepared fibrils were taken up by the cells in vitro upon dissolution.

### Aβ42 fibrils detection by immunofluorescence

Immunofluorescence studies demonstrated specific staining of Aβ42 fibrils (Fig. 2D) on SH-SY5Y cells. No staining was observed in SH-SY5Y cells in the absence of the Aβ42 fibrils (Fig. 2C; negative control (NC)).

### Microarray analysis
#### *Altered gene expression profiles in multiple comparisons between AD and treatment groups*

We obtained gene expression profiles using the Affymetrix Expression Console and Transcriptome Analysis Console (TAC) software from multiple comparison between four

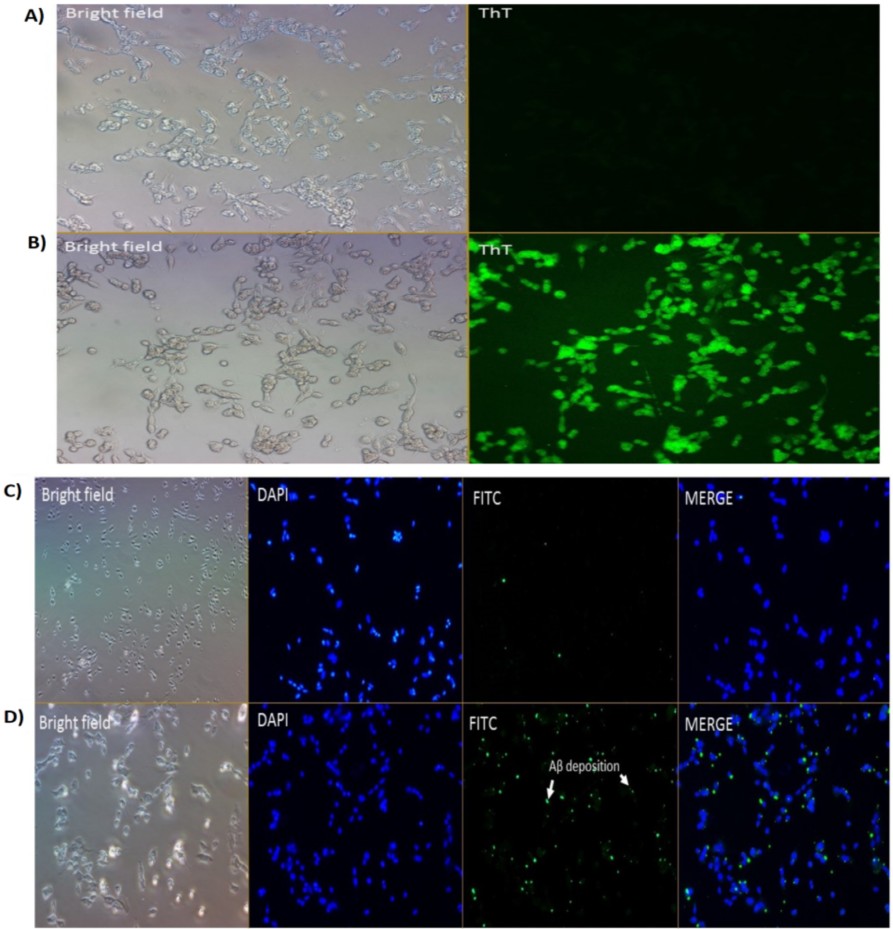

**Figure 2** Cell images of Aβ in SH-SY5Y cells under (A & B) Brightfield and Fluoresence Microscopy and (C & D) Immunofluorescence Microscopy under 20X magnification. (A) untreated SH-SY5Y cells and (B) SH-SY5Y cells treated using Thioflavin T (ThT) and (C) negative control and (D) treated cells (primary Ab: 1:200; secondary Ab: 1:1000 dilution). The stains 4′,6-diamidino-2-phenylindole (DAPI) was used to stain the nuclei (blue) and Fluorescein isothiocyanate (FITC) was used to stain the amyloid beta peptides (Aβ) (green). Scale bar = 50 μm.

groups of cells. These groups were (1) added with Aβ, which is identified as AD group or control (Aβ-C); (2) cells treated with curcumin for 24 h, followed by addition of Aβ −CuR + Aβ group; (3) cells treated with piperine, followed by addition of Aβ −Pip + Aβ group and (4) cells treated with combined curcumin and piperine, followed by addition of Aβ −CP + Aβ group. As shown in Fig. 3, all samples with no overlapped distribution in the Principal component analysis (PCA) exhibited clear separation between groups (Aβ −C; CuR + Aβ; Pip + Aβ and CP + Aβ) by hierarchical clustering of their expression profiles (Fig. 4). The mapping of expression values to intensities was depicted by color-bar created by the range of values in their respective conditions, i.e.-red for up-regulation and blue for down-regulation of genes.
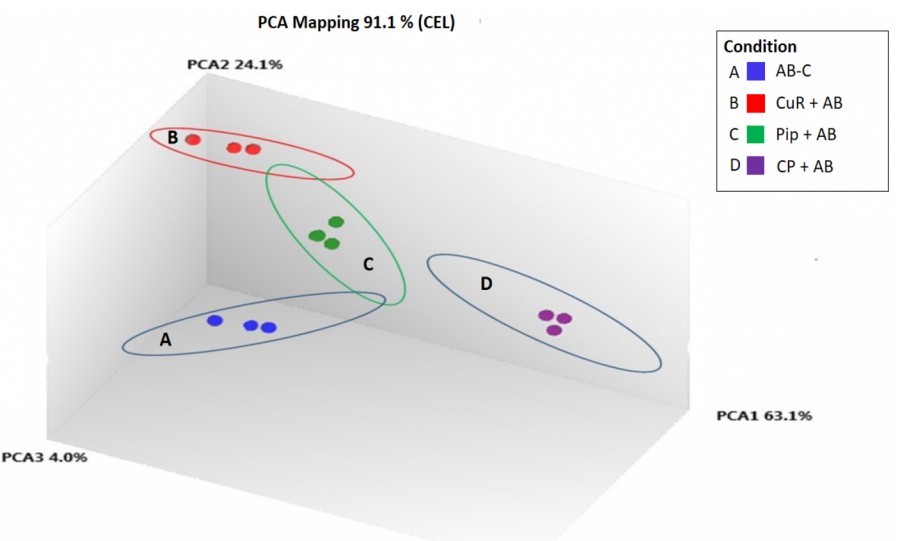

**Figure 3 Principal Component Analysis Mapping of SH-SY5Y cells under various conditions.** A-blue dots represent $A\beta$ Control; B-red dots represent CuR + $A\beta$; C-green dots represent Pip + $A\beta$ and; D-purple dots represent CP + $A\beta$. All samples with no overlapped distr. C, Control (no treatment); CuR, Curcumin; Pip, Piperine and CP, Combined curcumin and piperine.

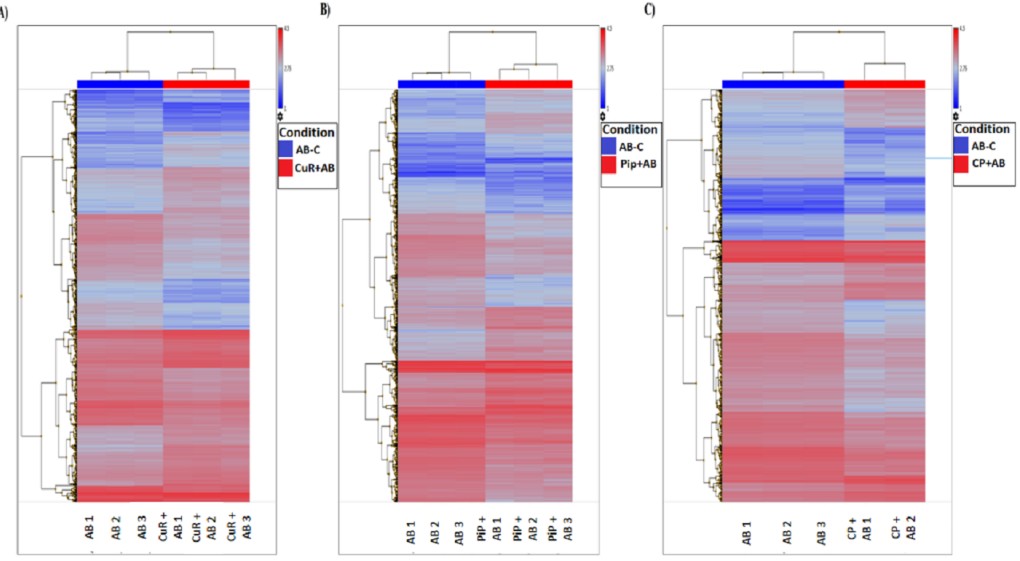

**Figure 4 Hierarchical Clustering of three different groups reveals clear separation of samples.** (A) [CuR + $A\beta$] vs. $A\beta$, (B) [Pip + $A\beta$] vs. $A\beta$ and (C) [CP + $A\beta$] vs. $A\beta$. (ANOVA: $P < 0.05$, log2> 6.64, a fold change $\geq 2$ or $\leq -2$) ($A\beta$, Amyloid beta; Curcumin; Pip, Piperine and CP, Combined curcumin and piperine).

**Table 1  Number of significant differentially expressed genes compared between treatment (curcumin and piperine singularly; combined curcumin and piperine) and control groups (Aβ).**

| Expression of genes | Total number of up and down regulated genes | | |
|---|---|---|---|
| | [CuR + Aβ] vs. Aβ | [Pip + Aβ] vs. Aβ | [CP + Aβ] vs. Aβ |
| Up-regulated | 1,723 | 2,028 | 1,210 |
| Down-regulated | 1,481 | 1,635 | 1,904 |
| Total genes | 3,204 | 3,663 | 3,114 |

By analyzing expression profiles of these samples using TAC software, we found that from 21,448 total number of genes altered, 9,951 genes passed the filter criteria where, 1,723 genes were up-regulated and 1481 genes were down-regulated in [CuR + Aβ] vs. Aβ group; whereas 2028 genes were up-regulated and 1,635 genes were down-regulated in [Pip + Aβ] vs. Aβ group; and 1,210 genes were up-regulated, 1,904 genes were down-regulated in [CP + Aβ] vs. Aβ group (filter criteria: ANOVA: $P < 0.05$, a lower bi-weight average signal (log2) >6.64, a fold change $\geq 2$ or $\leq -2$ (Table 1). Top 20 differentially expressed genes for each comparison are shown in Tables 2, 3 and 4.

### Individual treatment (CuR + Aβ vs. Aβ)

From total 1,723 upregulated and 1,481 downregulated genes differentially expressed in CuR + Aβ group, we presented the top 10 upregulated and top 10 downregulated genes that were significantly expressed (Table 2). We had identified the top 10 upregulated genes as *TGIF1, GABARAPL1, SYPL1, PICALMS, CTSB, IGFBP3, FSTL1, LBH, GABRA4* and *CASP8AP2,* while the top 10 downregulated genes were *ITGA9, AK5, SPRY1, APOE4, VIM, VAMP1, RAB5B, SCN3B, LFR5* and *LCOR.* We sought to evaluate the involvement of these genes in Aβ pathway and found that from literatures, i.e., *SYPL1* in synaptic modulation, *PICALM* in endothelial Aβ trafficking mechanism, *CTSB* in Aβ degrading enzyme, *APOE* in APP processing, *CASP8AP2* in apoptosis, *AK5* in neurite degeneration whereas *GABARAPL1 and RAB5B* were involved in macroautophagy.

### Individual treatment (Pip + Aβ vs. Aβ)

From the total 2,028 upregulated and 1635 downregulated genes differentially expressed in Pip + Aβ group, we presented the top 10 upregulated and top 10 downregulated genes that were significantly expressed in Table 3. We had identified the top 10 upregulated genes as *GABARAPL1, CUL3, CASP8AP2, AP3B2, CREB1, DCTD, GPC3, CTSB, FSTL1* and *GABRA4.* While the top 10 downregulated genes were *INA, RAB5B, OLFM1, HEATR5A, VAMP1, PSEN1, AK5, BMF, LAMP1* and *LYN.* We sought to evaluate the involvement of these genes in Aβ pathway and found that from literatures, *CREB1* involved in Ubiquitin proteasome system (UPS) pathway, *CTSB* in AB degrading enzyme, *PSEN1* in APP processing, *CASP8AP2* and *BMF* in apoptosis whereas *GABARAPL1 and RAB5B* are involved in macroautophagy.

### Combined treatment (CP + Aβ vs. Aβ)

From a total of 1,210 upregulated and 1904 downregulated genes differentially expressed in the CP + Aβ group, we presented the top 10 upregulated and top 10 downregulated

Abdul Manap et al. (2020), *PeerJ*, DOI 10.7717/peerj.10003

**Table 2  Top 10 most significant up-regulated and down-regulated genes between Aβ-SH-SY5Y cells treated with curcumin and untreated cells.**

| No. | Transcript ID | Top 10 up-regulated genes | Top 10 down-regulated genes | *P*-value | Fold-change | Entrez ID | Protein description |
|-----|---------------|---------------------------|-----------------------------|-----------|-------------|-----------|---------------------|
| 1 | TC1800006513.hg.1 | *TGIF1* | – | 1.07E−09 | 4.67 | 7050 | TGFB-induced factor homeobox 1 |
| 2. | TC1200006787.hg.1 | *GABARAPL1* | | 1.31E−09 | 5.29 | 23710 | GABA(A) receptor-associated protein like 1 |
| 3. | TC0700012180.hg.1 | *SYPL1* | | 4.56E−09 | 2.04 | 6856 | synaptophysin-like 1 |
| 4. | TC1100011838.hg.1 | *PICALM* | | 8.10E−09 | 5.04 | 8301 | phosphatidylinositol binding clathrin assembly protein |
| 5 | TC0800009619.hg.1 | *CTSB* | – | 1.82E−08 | 20.73 | 1508 | cathepsin B |
| 6 | TC0700010965.hg.1 | *IGFBP3* | – | 3.16E−08 | 5.01 | 3486 | insulin like growth factor binding protein 3 |
| 7 | TC0300012145.hg.1 | *FSTL1* | – | 3.38E−08 | 6.62 | 11167 | follistatin like 1; microRNA 198 |
| 8 | TC0200016424.hg.1 | *LBH* | – | 3.77E−08 | 16.14 | 81606 | limb bud and heart development |
| 9 | TC0400010602.hg.1 | *GABRA4* | – | 3.89E−08 | 12.58 | 2557 | gamma-aminobutyric acid (GABA) A receptor, alpha 4 |
| 10 | TC0600008780.hg.1 | *CASP8AP2* | – | 3.93E−08 | 3.29 | 9994 | caspase 8 associated protein 2 |
| 11 | TC0300007051.hg.1 | – | *ITGA9* | 4.31E−08 | −5.99 | 3680 | integrin alpha 9 |
| 12 | TC0100008797.hg.1 | | *AK5* | 1.01E−07 | −4.28 | 26289 | adenylate kinase 5 |
| 13 | TC0400008628.hg.1 | – | *SPRY1* | 2.23E−07 | −3.2 | 10252 | sprouty RTK signaling antagonist 1 |
| 14 | TC1900011758.hg.1 | – | *APOE4* | 3.15E−07 | −17.74 | 348 | apolipoprotein E |
| 15 | TC1000006891.hg.1 | – | *VIM* | 4.12E−07 | −3.81 | 7431 | vimentin |
| 16 | TC1200009734.hg.1 | – | *VAMP1* | 4.63E−07 | −4.25 | 6843 | vesicle associated membrane protein 1 |
| 17 | TC2000009915.hg.1 | – | *RAB5B* | 1.44E−06 | −2.23 | 55969 | Rab5-interacting protein family |
| 18 | TC1100012633.hg.1 | – | *SCN3B* | 1.50E−06 | −8.22 | 55800 | Immunoglobulin V-set domain |
| 19 | TC1200008176.hg.1 | – | *LGR5* | 1.61E−06 | −3.14 | 8549 | G protein-coupled receptor |
| 20 | TC1000008556.hg.1 | – | *LCOR* | 1.79E−06 | −4.49 | 84458 | ligand dependent nuclear receptor corepressor like |

**Notes.**

CuR,  Curcumin; Pip,  Piperine; CP,  Combined curcumin and piperine.

Abdul Manap et al. (2020), *PeerJ*, DOI 10.7717/peerj.10003

**Table 3** Top 10 most significant up-regulated and down-regulated genes between Aβ-SH-SY5Y cells treated with piperine and untreated cells.

| No. | Transcript ID | Top 10 up-regulated genes | Top 10 down-regulated genes | *P*-value | Fold-change | Entrez ID | Protein description |
|---|---|---|---|---|---|---|---|
| 1 | TC1200006787.hg.1 | GABARAPL1 | – | 2.87E−09 | 17.61 | 23710 | GABA(A) receptor-associated protein like 1 |
| 2 | TC0200015887.hg.1 | CUL3 | – | 4.02E−09 | 23.68 | 8452 | Cullin, N-terminal; Cullin protein |
| 3 | TC0600008780.hg.1 | CASP8AP2 | – | 9.73E−09 | 13.46 | 9994 | Caspase 8 associated protein 2 |
| 4 | TC1500010244.hg.1 | AP3B2 | – | 1.13E−08 | 25.45 | 8120 | Adaptor-related protein complex 3 |
| 5 | TC0200010607.hg.1 | CREB1 | – | 2.00E−08 | 31.86 | 1385 | cAMP responsive element binding protein 1 |
| 6 | TC0400012551.hg.1 | DCTD | – | 2.85E−08 | 6.00 | 1635 | Cytidine and deoxycytidylate deaminases |
| 7 | TC0X00010851.hg.1 | GPC3 | – | 3.39E−08 | 7.16 | 2719 | glypican 3 |
| 8 | TC0800009619.hg.1 | CTSB | – | 4.36E−08 | 17.33 | 1508 | cathepsin B |
| 9 | TC0300012145.hg.1 | FSTL1 | – | 5.71E−08 | 15.96 | 11167 | Osteonectin EGF domain |
| 10 | TC0400010602.hg.1 | GABRA4 | – | 6.04E−08 | 4.22 | 2557 | gamma-aminobutyric acid (GABA) A receptor, alpha 4 |
| 11 | TC1000008767.hg.1 | – | INA | 2.23E−08 | −10.07 | 9118 | internexin neuronal intermediate filament protein, alpha |
| 12 | TC2000009915.hg.1 | – | RAB5B | 3.45E−08 | −15.27 | 55969 | Rab5-interacting protein family |
| 13 | TC0900009145.hg.1 | – | OLFM1 | 3.72E−08 | −7.57 | 10439 | olfactomedin 1 |
| 14 | TC1400010728.hg.1 | – | HEATR5A | 8.10E−08 | −7.57 | 25938 | HEAT repeat containing 5A |
| 15 | TC1200009734.hg.1 | – | VAMP1 | 1.32E−07 | −4.29 | 6843 | vesicle associated membrane protein 1 |
| 16 | TC1400007628.hg.1 | – | PSEN1 | 1.34E−07 | −15.29 | 5663 | presenilin 1 |
| 17. | TC0100008797.hg.1 | | AK5 | 1.59E−07 | −15.36 | 26289 | adenylate kinase 5 |
| 18 | TC1500009082.hg.1 | – | BMF | 1.65E−07 | −5.07 | 90427 | Bcl2 modifying factor |
| 19 | TC1300008125.hg.1 | – | LAMP1 | 1.80E−07 | −5.24 | 7431 | lysosomal-associated membrane protein 1 |
| 20 | TC0800007688.hg.1 | – | LYN | 2.14E−07 | −17.16 | 4067 | LYN proto-oncogene |

**Table 4  Top 10 most significant up-regulated and down-regulated genes between Aβ-SH-SY5Y cells treated with curcumin-piperine in combination and untreated cells.**

| No. | Transcript ID | Top 10 up-regulated genes | Top 10 down-regulated genes | P-value | Fold-change | Entrez ID | Protein description |
|---|---|---|---|---|---|---|---|
| 1 | TC1200006787.hg.1 | GABARAPL1 | | 1.06E−07 | 18.72 | 23710 | GABA(A) receptor-associated protein like 1 |
| 2 | TC1100008262.hg.1 | FADD | – | 1.11E−07 | 22.91 | 8772 | Fas (TNFRSF6)-associated via death domain |
| 3 | TC1200007861.hg.1 | LRP1 | – | 1.64E−07 | 5.77 | 4035 | LDL receptor related protein 1 |
| 4 | TC2000007900.hg.1 | VAPB | – | 1.97E−07 | 4.61 | 9217 | VAMP (vesicle-associated membrane protein)-associated protein B and C |
| 5 | TC1500010244.hg.1 | AP3B2 | – | 2.21E−07 | 4.49 | 8120 | Adaptor-related protein complex 3 |
| 6 | TC0800009619.hg.1 | CTSB | – | 2.22E−07 | 3.65 | 1508 | cathepsin B |
| 7 | TC0200015887.hg.1 | CUL3 | | 2.72E−07 | 3.53 | 8452 | Cullin, N-terminal; Cullin protein |
| 8 | TC2100007822.hg.1 | ADAMTS5 | – | 1.97E−07 | 5.08 | 11096 | ADAM metallopeptidase with thrombospondin type 1 motif 5 |
| 9 | TC1500010350.hg.1 | NTRK3 | – | 3.43E−07 | 5.73 | 4916 | neurotrophic tyrosine kinase, receptor, type 3 |
| 10 | TC1800006513.hg.1 | TGIF1 | – | 3.88E−07 | 16.28 | 7050 | TGFB-induced factor homeobox 1 |
| 11 | TC2000006674.hg.1 | – | PLCB4 | 2.89E−07 | −3.40 | 5332 | phospholipase C, beta 4 |
| 12 | TC1200008176.hg.1 | – | LGR5 | 3.76E−07 | −5.16 | 8549 | leucine-rich repeat containing G protein-coupled receptor 5 |
| 13 | TC0600010960.hg.1 | – | TPMT | 3.99E−07 | −13.43 | 7172 | thiopurine S-methyltransferase |
| 14 | TC1000006891.hg.1 | – | VIM | 4.77E−07 | −5.42 | 7431 | vimentin |
| 15 | TC0900009145.hg.1 | | OLFM1 | 5.50E−07 | −2.37 | 10439 | olfactomedin 1 |
| 16 | TC0M00006440.hg.1 | – | ND3 | 5.68E−07 | −2.54 | NA | NADH dehydrogenase, subunit 3 (complex I) |
| 17 | TC2000009915.hg.1 | – | RAB5B | 7.15E−07 | −10.21 | 55969 | Rab5-interacting protein family |
| 18 | TC0100008797.hg.1 | | AK5 | 8.78E−07 | −14.69 | 26289 | adenylate kinase 5 |
| 19 | TC0800007688.hg.1 | – | LYN | 1.03E−06 | −6.87 | 4067 | LYN proto-oncogene |
| 20 | TC1000008767.hg.1 | – | INA | 1.17E−06 | −3.88 | 9118 | internexin neuronal intermediate filament protein, alpha |
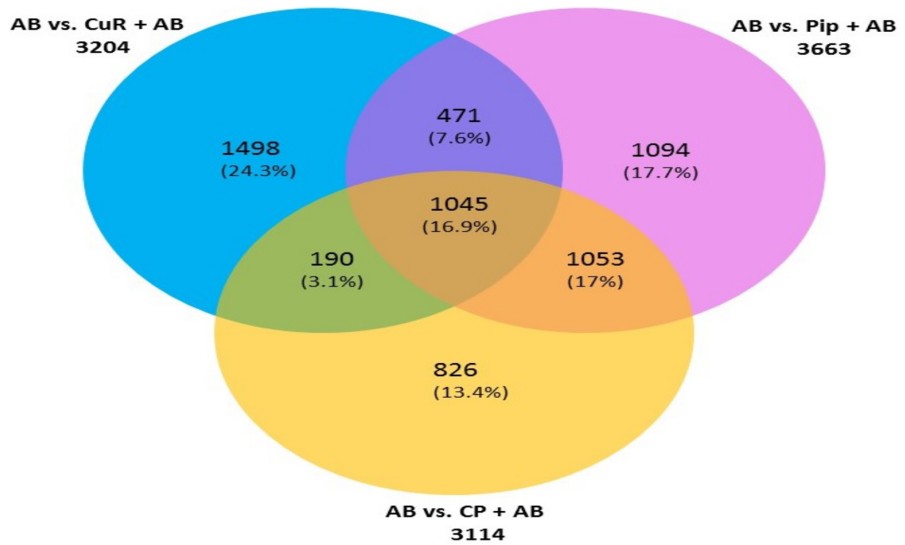

**Figure 5** **Venn diagram showing overlapping genes with significant gene expression in each comparison.** Total number of up- and downregulated genes in each group are shown in parentheses (CuR, Curcumin; Pip, Piperine and CP, Combined curcumin and piperine). (ANOVA: $P < 0.05$, log2 > 6.64, a fold change $\geq 2$ or $\leq -2$).

genes that were significantly expressed in Table 4. We had identified the top 10 upregulated genes as *GABARAPL1, FADD, LRP1, VAPB, AP3B2, CTSB, CUL3, ADAMTS5, NTRK3* and *TGIF1*, while the top 10 downregulated genes were *PLCB4, LRG5, TPMT, VIM, OLFM1, ND3, RAB5, AK5 LYN* and *INA*. We sought to evaluate the involvement of these genes in A$\beta$ pathway and found that from literatures, *ADAMTS5* an*d CTSB* were involved in A$\beta$ degrading enzyme, *LRP1* in endothelial A$\beta$ trafficking, *FADD* in apoptosis, *AK5* in neurite degeneration whereas *GABARAPL1 and RAB5B* were involved in macroautophagy.

We then compared the genes among the three groups, and found that all groups shared a total of 1,045 genes (16.9%) that were differentially altered in the comparison between AD group (A$\beta$-C) and individual (curcumin and piperine) as well as in combined treatment (combined curcumin and piperine) (Fig. 5). A top 20 differentially expressed genes were shared in these three groups as shown in Table 5. Protein-protein interaction network was performed on the top 20 (Fig. 6A) significant differentially expressed genes (DEGs) that are listed in Table 5 by using Network Analyst Software 3.0. While Table 6 showed the top 20 most significant pathways that are involved and potentially involved in AD.

### The expression level of commonly altered genes in treatment and control groups increase Alzheimer progression through Alzheimer's disease pathway

Among the total genes significantly differentially expressed in each group (3204 genes in [CuR + A$\beta$] vs. A$\beta$, 3663 genes in [Pip + A$\beta$] vs. A$\beta$ and 3114 genes [CP + A$\beta$] vs. A$\beta$), only 25 genes have been identified to be involved in AD pathway, regardless of A$\beta$ specific pathways, characterized by WikiPathways (Table 7). However, only 6 genes were shared among the three comparisons identified as *APOE, FADD, LRP1, PLCB3, PLCB4 and*

**Table 5**  Top 20 most significant differentially expressed genes in all three groups (CuR + Aβ vs. Aβ; Pip + Aβ vs. Aβ & CP + Aβ vs. Aβ).

| Transcript ID | Gene symbol | Description | Entrez ID |
|---|---|---|---|
| TC0100008797.hg.1 | AK5 | adenylate kinase 5 | 26289 |
| TC0100015895.hg.1 | RAB13 | RAB13, member RAS oncogene family | 5872 |
| TC0200011975.hg.1 | TP53I3 | tumor protein p53 inducible protein 3 | 9540 |
| TC1000012169.hg.1 | ADAM12 | ADAM metallopeptidase domain 12 | 8038 |
| TC1100008262.hg.1 | FADD | Fas (TNFRSF6)-associated via death domain | 8772 |
| TC1200006787.hg.1 | GABARAPL1 | GABA(A) receptor-associated protein like 1 | 23710 |
| TC1600009202.hg.1 | CREBBP | CREB binding protein | 1387 |
| TSUnmapped00000228.hg.1 | NDUFA10 | NADH dehydrogenase (ubiquinone) 1 alpha subcomplex, 10, 42kDa | 4705 |
| TC0100014349.hg.1 | JUN | jun proto-oncogene | 3725 |
| TC0200007096.hg.1 | FOSL2 | FOS-like antigen 2 | 2355 |
| TC0100007789.hg.1 | AGO3 | argonaute RISC catalytic component 3 | 192669 |
| TC0100008243.hg.1 | ELAVL4 | ELAV like neuron-specific RNA binding protein 4 | 1996 |
| TC0100008517.hg.1 | NFIA | nuclear factor I/A | 4774 |
| TC0100008554.hg.1 | USP1 | ubiquitin specific peptidase 1 | 7398 |
| TC0100008664.hg.1 | GADD45A | growth arrest and DNA-damage-inducible, alpha | 1647 |
| TC0100008692.hg.1 | SRSF11 | serine/arginine-rich splicing factor 11 | 9295 |
| TC0100008845.hg.1 | ADGRL2 | adhesion G protein-coupled receptor L2 | 23266 |
| TC0100008912.hg.1 | CYR61 | cysteine-rich, angiogenic inducer, 61 | 3491 |
| TC0100008938.hg.1 | LMO4 | LIM domain only 4 | 8543 |
| TC0100009020.hg.1 | CDC7 | cell division cycle 7 | 8317 |

**Notes.**
CuR, Curcumin; Pip, Piperine; CP, Combined curcumin and piperine.

*GRIN2A* (shown in bold). *APOE, FADD, LRP1 and PLCB3* appeared to be up-regulated in all three comparisons while *PLCB4 and GRIN2A* were downregulated. Protein-protein interaction network was performed on top 25 (Fig. 6B) and top 8 (Fig. 6C) significantly differentially expressed genes (DEGs) that are listed in Table 7 by using Network Analyst Software 3.0.

### Gene expression changes validated by qPCR

The results obtained from qPCR confirmed the expression changes of the selected genes from microarray (Fig. 7).

## DISCUSSION

The present study was performed to assess the anti-amyloidogenic role of individual and combined curcumin and piperine. Thioflavin T microscopy staining and immunofluorescence confirmed the uptake of Aβ in SH-SY5Y cells. From the microarray analysis, we demonstrated for the first time that these compounds exhibited their actions against Aβ via modulation of various mechanistic pathways that are responsible either via production or clearance of Aβ. In addition, it has been shown that the neuroprotective effects of these compounds on degeneration have been induced by Aβ such as synaptic impairment, degradation of the ubiquitin proteasome system (UPS), apoptosis, and neurite
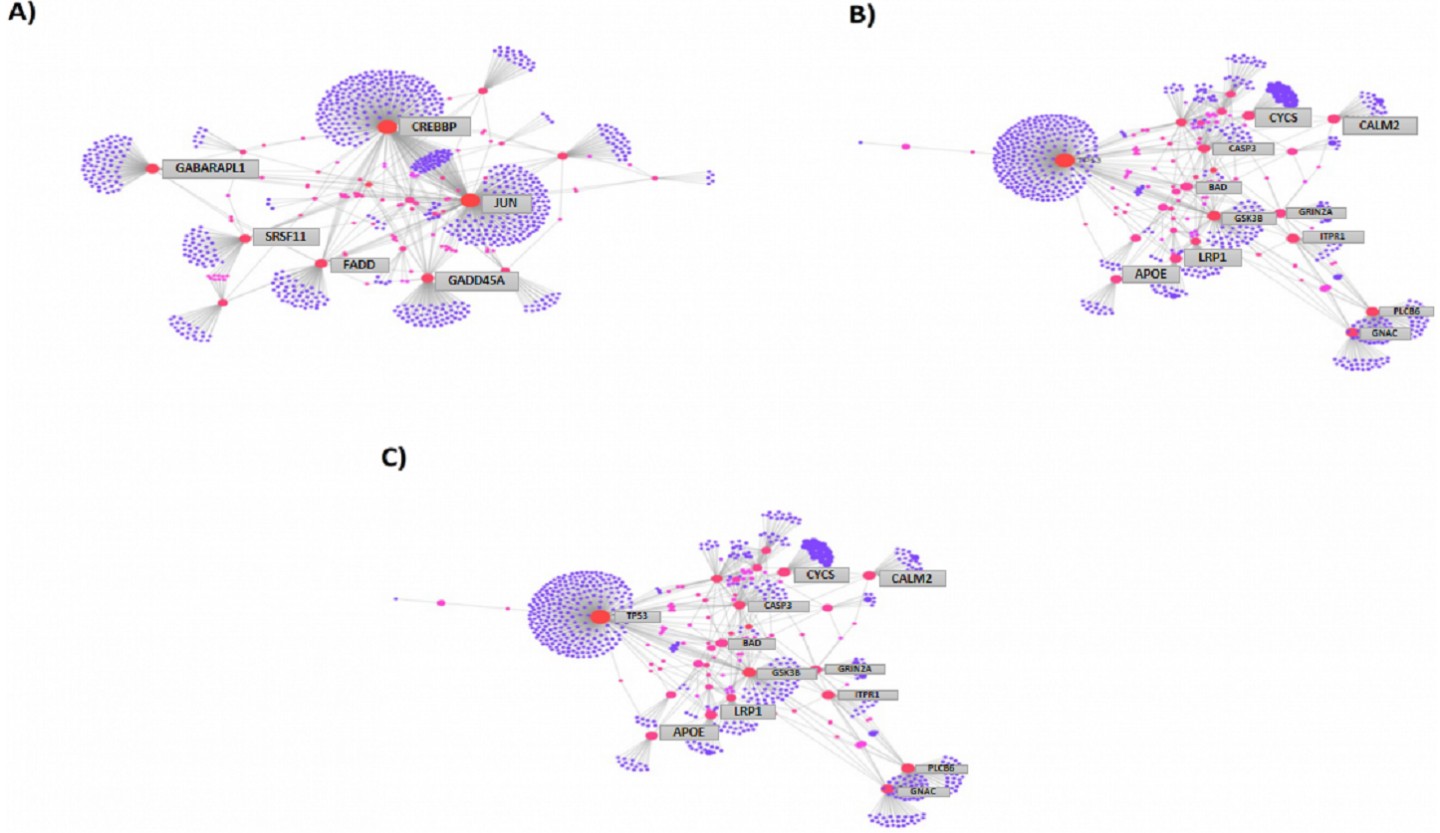

**Figure 6** **Protein-protein interaction network of differentially expressed genes (DEGs) by using Network Analyst Software 3.0.** (A) On the top 20 most significant DEG, (B) top 25 most significantly DEG and (C) top eight most significantly DEG. The color represents the degree of the nodes. Red nodes are most important interactions, followed by pink nodes and finally purple nodes. Nodes in blue represent those proteins interacting in cancer pathways.

degeneration as discussed in the following sections . Protein-protein interaction (PPI) has shown that various networks of interactions tend to be potentially involved in AD. Although there are some limitations of microarray data which often hinder data interpretation, the key advantage of the microarray provides a new method for tackling ambiguity, enabling simultaneous overviews of several cellular pathways.

## Modulation of macroautophagy process was observed in all single and combined treatment of curcumin and piperine

A key determinant of cell survival and longevity is macroautophagy, a lysosomal pathway for organelles turnover and long-lived proteins. Previous study by *Yu et al. (2005)* showed that, neuronal macroautophagy was induced early in AD and right before A$\beta$ deposits extracellularly in the mouse model of presenilin (PS) 1/A$\beta$ precursor protein pathology/pathway (APP) (*Yu et al., 2005*). From our findings, we found that expression of *GABARAP* and *RAB5* genes were altered in all three groups indicating protective roles of the compounds against A$\beta$ toxicities, by modulating the macroautophagy process.

**Table 6** Shows top 20 most significant pathway ($p < 0.05$) appeared in 1,045 genes that are involved or potentially involved in Alzheimer Disease.

| Pathway | Total | Expected | Hits | *P*.Value | FDR |
|---|---|---|---|---|---|
| Alzheimer's disease | 171 | 12.7 | 74 | 3.22E−41 | 1.02E−38 |
| Long-term potentiation | 67 | 4.99 | 74 | 2.18E−36 | 3.46E−34 |
| TGF-beta signaling pathway | 92 | 6.85 | 124 | 1.47E−33 | 1.56E−31 |
| Dopaminergic synapse | 131 | 9.75 | 52 | 3.58E−30 | 2.84E−28 |
| Choline metabolism in cancer | 99 | 7.37 | 69 | 2.96E−29 | 1.88E−27 |
| Basal transcription factors | 45 | 3.35 | 56 | 3.62E−28 | 1.92E−26 |
| Basal cell carcinoma | 63 | 4.69 | 69 | 1.11E−26 | 5.02E−25 |
| Cytokine-cytokine receptor interaction | 294 | 21.9 | 50 | 4.42E−25 | 1.76E−23 |
| cAMP signaling pathway | 212 | 15.8 | 43 | 1.30E−24 | 4.61E−23 |
| Apoptosis - multiple species | 33 | 2.46 | 62 | 1.79E−23 | 5.69E−22 |
| Insulin signaling pathway | 137 | 10.2 | 54 | 2.21E−23 | 6.39E−22 |
| Ras signaling pathway | 232 | 17.3 | 58 | 2.86E−22 | 7.59E−21 |
| Ubiquitin mediated proteolysis | 137 | 10.2 | 42 | 2.97E−21 | 7.27E−20 |
| Adipocytokine signaling pathway | 69 | 5.14 | 41 | 1.72E−20 | 3.90E−19 |
| B cell receptor signaling pathway | 71 | 5.29 | 33 | 1.80E−19 | 3.82E−18 |
| Autophagy | 128 | 9.53 | 51 | 3.07E−19 | 6.11E−18 |
| p53 signaling pathway | 72 | 5.36 | 34 | 4.05E−19 | 7.58E−18 |
| NF-kappa B signaling pathway | 100 | 7.44 | 44 | 5.76E−19 | 1.02E−17 |
| FoxO signaling pathway | 132 | 9.83 | 52 | 1.20E−18 | 2.01E−17 |
| AGE-RAGE signaling pathway in diabetic complications | 100 | 7.44 | 45 | 3.35E−18 | 5.32E−17 |

Gamma-aminobutyric acid receptor-associated proteins (*GABARAPs; GABARAP, GABARAP-L1, GABARAP-L2*), ubiquitin-like proteins that are covalently conjugated to phosphatidylethanolamine (PE) on autophagosomal membranes promote the formation, elongation and maturation of autophagosomes (*Kabeya et al., 2004*; *Nakatogawa, 2013*). This *GABA* receptor modulators have been explored in AD as a prospective therapeutic approach. Our data indicates that the upregulation of *GABARAPL1* was caused by the treatment of curcumin, piperine singularly and combination of these (Fig. 8), whereas it was downregulated in the control group (A$\beta$ without any treatment). These findings are in agreement with earlier reports on the effects of A$\beta$ at inhibitory synapses. *Ulrich (2015)* investigated the impact of acute A$\beta$1-42 application on GABAergic synaptic transmission in rat somatosensory cortex in vitro (*Ulrich, 2015*). He found in his study that, with intracellular applications of p4, a blocker of internalization of the GABA(A) receptor, the A$\beta$-induced IPSC reduction could be avoided, which may conclude that A$\beta$ weakens synaptic inhibition via downregulation of GABA(A) receptors (*Ulrich, 2015*). Moreover, *GABA* also found to suppress uptake of A$\beta$ in neurons via the receptor for advanced glycation of end-products (*Sun et al., 2012*). *RAB5* is a member of the RAS oncogene family (*Nakhaei-Rad et al., 2018*). The small GTPases Rab are important intracellular membrane trafficking regulators, ranging from transport vesicles to membrane fusion*RAB5* endosomes are the main sites for $\beta$-secretase (*Grbovic et al., 2003*). Sustained *RAB5* activation promotes

**Table 7  25 differentially expressed genes involved in AD pathway.**

| No. | Gene | Role in the pathogenesis of AD | Expression of Genes (up/down) | | |
|---|---|---|---|---|---|
| | | | CuR+A$\beta$ vs. AB | Pip+A$\beta$ vs. AB | CP+A$\beta$ vs. AB |
| 1 | *MME* | Most important A$\beta$-degrading enzymes (*Miners et al., 2012*) | up | up | – |
| 2 | **APOE** | **APOE2 carriers have a protective effect relative to APOE3 and APOE4 carriers, and therefore the APOE4 protein appears to be 'toxic' and more likely develop AD (Safieh, Korczyn & Michaelson, 2019)** | **up** | **up** | **up** |
| 3 | *NCSTN* | One of the $\gamma$-secretase genes. Mutations have been reported which linked with A $\beta$ formation (*Pink et al., 2013*) | up | – | – |
| 4 | *BAD* | Increased expression was observed in AD (*Kitamura et al., 1998*; *Ribarič & Ribarič, 2019*) | up | – | – |
| 5 | *TNFRSF1A* | Regulation of APP processing; genetic deletion of the TNF receptor gene *TNFRSF1A* in the APP 23 transgenic mouse model reduced both the number of amyloid plaques and the cognitive deficits in these mice (*He et al., 2007*; *McAlpine & Tansey, 2008*) | up | – | – |
| 6 | **FADD** | **Cortical FADD was lower in subjects with dementia and lower FADD was associated with a greater load of amyloid-$\beta$ pathology (Ramos-Miguel et al., 2017)** | **up** | **up** | **up** |
| 7 | **LRP1** | **LRP1 is an important mediator for the rapid removal of A$\beta$ from brain via transport across the blood–brain barrier (BBB) (Storck & Pietrzik, 2017)** | **up** | **up** | **up** |
| 8 | *CASP8* | Involved in amyloid processing (*Rehker et al., 2017*) | up | – | – |
| 9 | **PLCB3** | **Plays an important role in initiating receptor-mediated signal transduction (Lagercrantz et al., 1995). Limited finding in Alzheimer.** | **up** | **up** | **up** |
| 10 | *CAPN1* | Upregulation of calpain activation in the brain of AD activates CDK5, activates BACE1,therefore increase A $\beta$40 and A $\beta$42 production in transgenic mice (*Wen et al., 2008*) | up | – | – |
| 11 | *TP53* | The increase in the level of p53 has been detected in the brain tissue of AD patients (*Sajan et al., 2007*) | up | down | – |
| 12 | **PLCB4** | **Plays an important role in initiating receptor-mediated signal transduction (Lagercrantz et al. 1995). Limited literatures in Alzheimer.** | **down** | **down** | **down** |
| 13 | *GNAQ* | The expression level of Gnaq in SAMP8 mouse forebrain cortex was significantly decreased with age, alluding to the possibility that Gnaq expression may be closely associated with brain aging (*Chen et al., 2010*) | down | – | – |
| 14 | **GRIN2A** | **One of the NMDAR subunit gene. A missense mutation in the coding regions of the GRIN2B was found only in the brains of AD patients (Andreoli et al., 2014) while GRIN2A mutation of substitution p.N615K is found in a girl with early-onset epileptic encephalopathy (Endele et al., 2010).** | **down** | **down** | **down** |
| 15 | *ITPR1* | Involve in calcium signaling pathway. Mutations in this gene cause spinocerebellar ataxia (*Hisatsune & Mikoshiba, 2017*) | down | – | – |
**Table 7** (*continued*)

| No. | Gene | Role in the pathogenesis of AD | Expression of Genes (up/down) | | |
|-----|------|-------------------------------|---------------------|-------------------|------------------|
| | | | CuR+A$\beta$ vs. AB | Pip+A$\beta$ vs. AB | CP+A$\beta$ vs. AB |
| 16 | GSK3B | GSK3 activity and/or protein levels are increased in afflicted individuals with AD (*Hooper, Killick & Lovestone, 2008*) | down | – | – |
| 17 | PSEN1 | Presenilin 1 (PSEN1) encodes the catalytic subunit of $\gamma$-secretase, and PSEN1 mutations are the most common cause of early onset familial Alzheimer's disease (FAD) (*Sproul et al., 2014*) | – | up | – |
| 18 | PPP3CC | Referred to as calcineurin. For memory-associated disorder, AD, average levels of calcineurin expression and calcineurin activity for AD brains are decreased (*Gong et al., 2006*) | – | up | – |
| 19 | CASP9 | This gene has been reported to involve in neuroinflammation and apoptosis leading to the onset of AD (*Abid, Naseer & Kim, 2019*). | – | up | – |
| 20 | ERN1 | IRE1 impairment completely restored AD mice's learning and memory capacity, combined with enhanced synaptic function and increased long-term potential (LTP) (*Duran-Aniotz et al., 2017*). | – | up | – |
| 21 | IDE | Previously reported as a late-onset AD gene based on its potential to degrade amyloid $\beta$-protein (*Vepsäläinen et al., 2007*) | – | down | – |
| 22 | APH1A | Encodes a gamma secretase complex component. Polymorphisms were associated with an increased risk of developing sporadic Alzheimer's disease in a promoter region of this gene (*Wang & Jia, 2009*). | – | down | – |
| 23 | CYCS | Involve in mitochondrial dysfunction as well as inflammation and apoptosis linked to AD (*Kim et al., 2018*). | – | down | – |
| 24 | CALM2 | Down-expression of this gene from AD contribution was reported in the cerebellum of autistic patients (*Zeidán-Chuliá et al., 2014*) | – | – | down |
| 25 | CASP3 | This gene has been reported to involve in neuroinflammation and apoptosis leading to the onset of AD (*Abid, Naseer & Kim, 2019*). | – | – | down |

**Notes.**
Six genes were shared among the three comparisons identified as APOE, FADD, LRP1, PLCB3, PLCB4 and GRIN2A (shown in bold).

APP cleavage and builds toxic beta-CTFs ($\beta$CTFs) and A$\beta$ species. In turn, intense $\beta$CTFs and A$\beta$ also enhance active types of *RAB5*, leading in enlarged endosomes and accelerated amyloidogenic APP processing (*Nixon, 2017*). Downregulation of this *RAB5* by curcumin, piperine and CP group may indicate crucial inhibition of $\beta$-secretase activity by declining APP cleavage and preventing buildup of toxic A$\beta$ (Fig. 8).

## Modulation of endothelial A$\beta$ trafficking by PICALM and LRP1 through transcytosis

*PICALM* has been remarkable as robustly validated genetic risk factor for AD. *PICALM* is among the highly abundant clathrin adaptors in clathrin-coated endocytic vesicles and regulates endocytic processes in presynaptic active neuronal zones (*Blondeau et al., 2004*; *Koo et al., 2011*). *PICALM* regulates the formation of A$\beta$ through endocytosis of APP and $\gamma$-secretase, presumably in neurons (Fig. 9). Other scientists also noted that

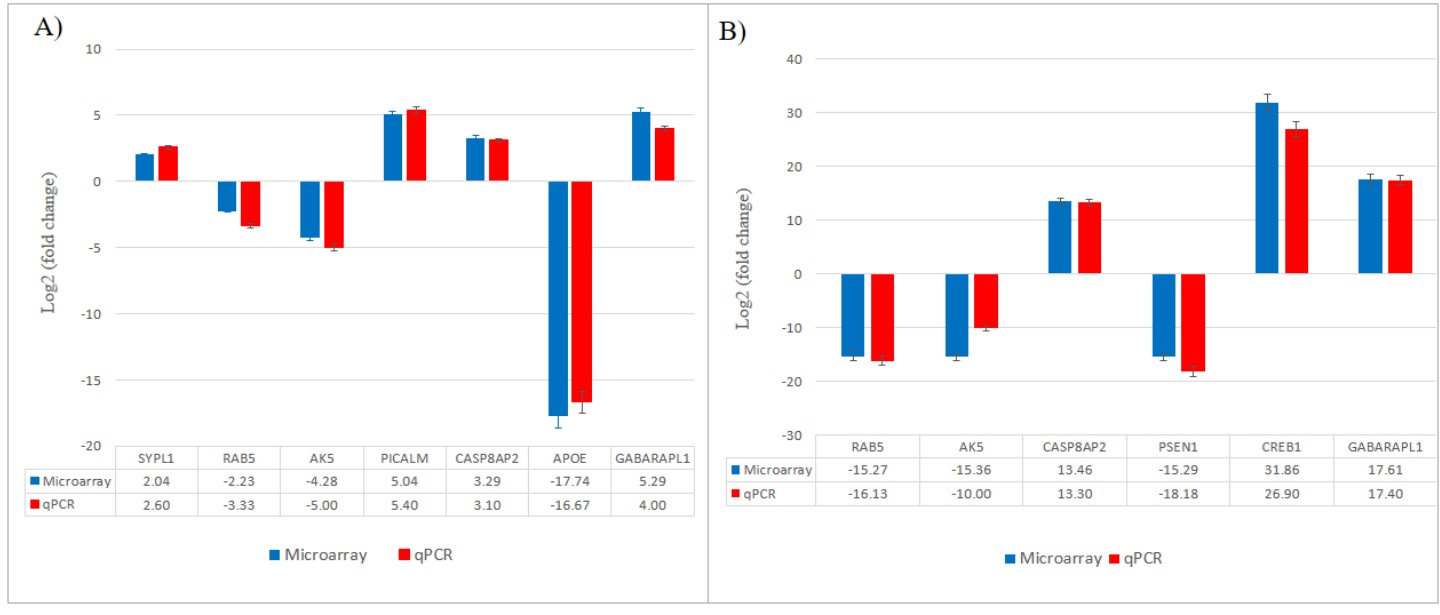

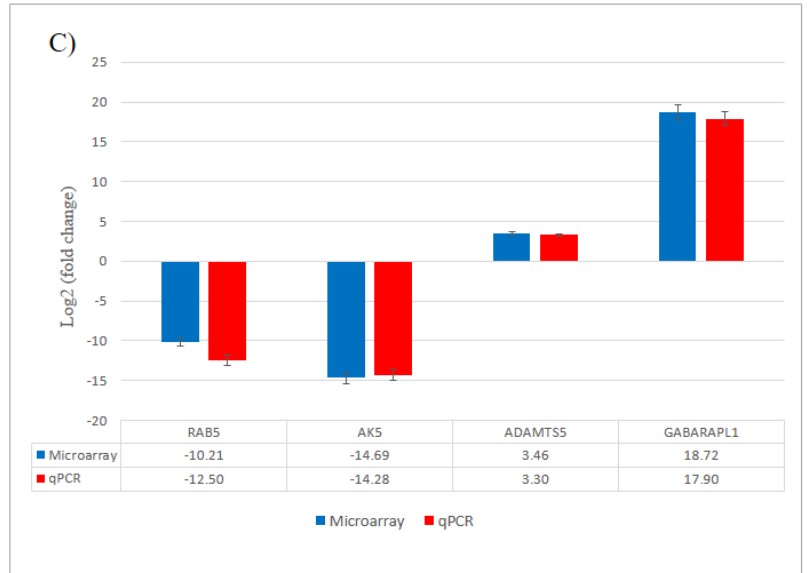

**Figure 7 Validation of selected differentially expressed gene candidates by qPCR in SH-SY5Y cells.** (A) CuR + Aβ; (B) Pip + Aβ; (C) CP + Aβ. Aβ, Amyloid beta; CuR, Curcumin; Pip, Piperine; CP, Curcumin-Piperine.

*PICALM* plays a significant role in tau clearance and autophagy, implying that *PICALM* is a multifunctional protein (*Moreau et al., 2015*). We found in our study that *PICALM* expression level was downregulated in Aβ group while upregulated in the group treated with curcumin. While *PICALM* expression was not observed in the group treated with piperine and in CP (Fig. 8). Our finding coincides with the previously reported studies where reduced expression of *PICALM* was observed in AD and murine brain endothelium associated Aβ pathology and cognitive decline (*Zhao et al., 2015*). In addition, they found

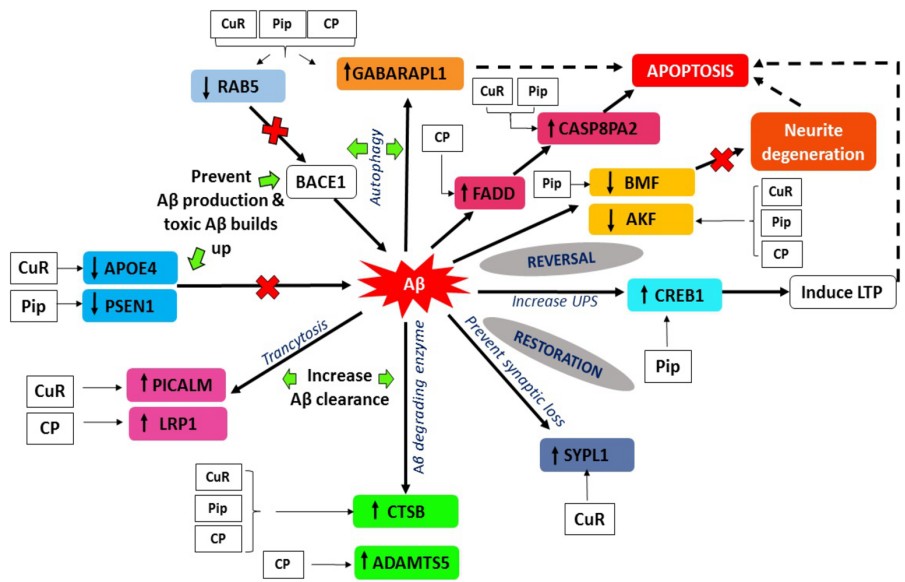

**Figure 8  Aβ extrinsic pathway and the genes observed in array analysis.** The genes shown were modulated either by singular or combined treatment of curcumin and piperine. (CuR, Curcumin; Pip, Piperine and CP, Combined curcumin and piperine).

that reduced *PICALM* level impaired Aβ clearance across the murine blood–brain barrier (BBB) and enhanced Aβ pathology in a way reversible through endothelial re-expression of *PICALM* (*Zhao et al., 2015*). Furthermore, we demonstrated that the expression level of the *LRP1* gene was upregulated in the group treated with CP (Fig. 8). The low-density lipoprotein related protein (*LRP*), a constituent of the low-density lipoprotein receptor (*LDLR*) family, is a multi-ligand receptor of which its physiological functions are performed by ligand endocytosis and by activation of multiple signal transduction pathways (*Herz & Strickland, 2001*). Previous study reported that extracellular Aβ must bind to low-density lipoprotein-related protein 1 (*LRP1*) in capillary endothelial cells of the brain to be transported through the cell to the bloodstream (*Deane et al., 2004*). Another study by *Zhao et al. (2015)* reported that *PICALM* was attached to the Aβ/*LRP1* complex within 30 s of the addition of Aβ to the basolateral membrane (*Zhao et al., 2015*). Upregulation of both *PICALM* and *LRP1* in our treatment groups may provide a significant finding on the modulatory approach of these compounds on the degeneration of toxic Aβ via facilitating transcytosis of Aβ.

## Disruption in Aβ production or rapid Aβ clearance mechanistic pathways against Aβ degeneration

The fundamental strategy to degenerate toxic Aβ was to stop, inhibit or disrupt the production of Aβ at an early point, before Aβ has been circulated into the neurons. Thus, the detrimental effects of the Aβ can be abolished at the early onset of the disease. Otherwise, an alternative strategy was to facilitate Aβ clearance through BBB, thus flushing off this toxic Aβ protein. We showed from our study that, curcumin and piperine were

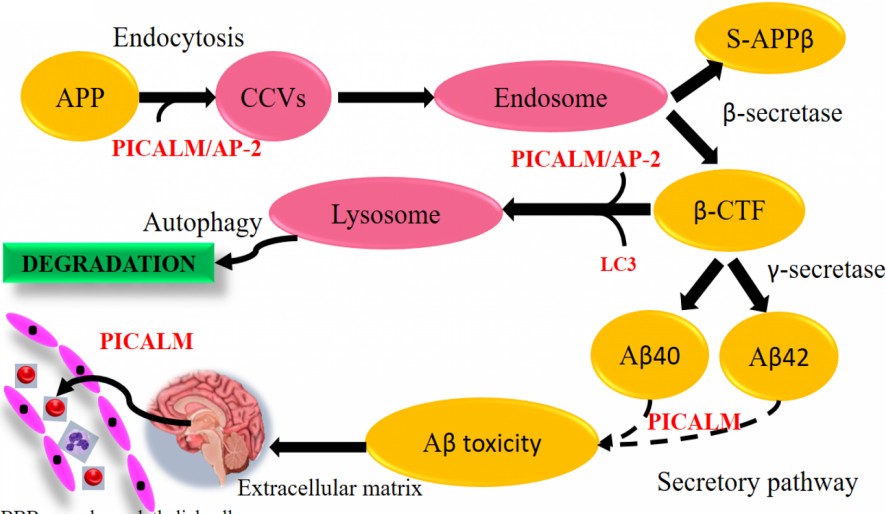

**Figure 9** **Aβ-dependent role of *PICALM* in AD. (Adapted from *Xu, Tan & Yu, 2015*).** *PICALM* can promote not only the formation of Aβ peptide through the endocytosis system, but also its clearance through the activation of autophagic APP-CTF processes and the facilitation of extracellular Aβ to cross the blood–brain barrier (BBB) vascular endothelial cells. (CCVS, clathrin-coated vesicles; LC3, light chain 3).

able to interfere with either of these two pathways-at the initial or end stages, through modulation of the expression of genes that are responsible at various stages of the pathway.

(i) *Upregulation* of *Aβ degrading enzyme*

Over the past century, enzyme-mediated degradation of Aβ has gained much attention. Many enzymes are capable of cleaving full-length Aβ in vitro, generating fragments that are generally less neurotoxic, less likely to aggregate and more easily cleared than Aβ in full length (*Hu et al., 2001*). We demonstrated that, *CTSB* gene, which produces Aβ degrading protease (*Miners et al., 2011*) was downregulated in Aβ group and upregulated in all treatment groups (Fig. 8). This gene encodes a member of the peptidase family C1. Alternative splicing of this gene results in various transcript forms (*Jevnikar & Kos, 2009*). Previous research showed that CTSB A $\beta_{1-42}$ cleavage produces C-terminally truncated peptides (Aβ1-40, Aβ $_{1-38}$, and Aβ $_{1-33}$), all of which are less toxic and less fibrillogenic than full-length Aβ $_{1-42}$ (*Mueller-Steiner et al., 2006*). The upregulation of this gene observed in our experimental data may indicate favorable mechanism of these compounds on *CTSB* gene expression by increasing the activity of the enzyme and elevating clearance of Aβ, thus provide therapeutic potential in AD.

On the other hand, we also observed that, besides the *CTSB* gene, *ADAMTS5* gene was also upregulated in the group treated with combined curcumin-piperine (Fig. 8). The expression of this gene did not appear in the group treated with curcumin and piperine singularly. The proteins A-like disintegrin and metalloproteinase (MMP) with motifs of thrombospondin (*ADAMTS*) were recognized as secreted protease enzymes, some of which may bind to the extracellular matrix (ECM) (*Porter et al., 2005*). The

crucial substrates of the enzymes are the aggregating chondroitin sulphate proteoglycans (CSPGs), including brevican, versican and aggrecan, recognized as the CNS's complete integral parts of the ECM (*Porter et al., 2005*). *ADAMTS* expression was discovered in the central nervous system (CNS) after various extensive research and is known to change in disease circumstances (*Miguel, Pollak & Lubec, 2005*; *Haddock et al., 2006*; *Gottschall & Howell, 2015*). The downregulation of *ADAMTS5* expression level in our Aβ group data was supported by previous study which demonstrated that *ADAMTS4* and *ADAMTS5* expressions were slightly under-expressed in case of AD which indicated the deficiency in the elimination of ECM in patients, which ultimately resulted in the accumulation of undesirable ECM compounds over time (*Pehlivan et al., 2016*). Likewise, increase expression of *ADAMTS5* from the combined treatment may assist in elevating the deterioration of ECM constituents, including Aβ senile plaque (*Pekny & Nilsson, 2005*), which in turn, might aid recovery by the elimination of the CSPGs.

(ii) *Inhibition of neurotoxic Aβ aggregation and plaque deposition*

Apolipoprotein E (*APOE*) gene polymorphism is a significant risk determinant for late-onset AD in patients, whose symptoms develop only after age 65 (*Lambert et al., 2013*). Of the three main types of *APOE*, *APOE⋆ ε4* is correlated with an increased risk (*Stocker et al., 2018*) and *APOE⋆ ε2* is linked with a reduced risk (*Escott-Price et al., 2019*) of AD compared to the common *APOE⋆ ε3* allele. Assembling proof indicates that the isoform *APOE⋆ ε4* drives amyloid pathology and impairs various aspects of normal brain function, increasing the risk of AD (*Fernandez et al., 2019*; *Safieh, Korczyn & Michaelson, 2019*). We demonstrated that APOE gene expression was upregulated in the group treated with Aβ alone, which may increase the risk for AD. Parallel to our findings, it was reported previously that HEK293 cell-derived *APOE* induced transcription of APP and generation of Aβ in human embryonic stem cells and iPSC-derived neurons (*Huang et al., 2018*). The effects of this is depended on isoforms, with *APOE⋆ ε4* more profoundly accelerating Aβ production than other isoforms (*Huang et al., 2018*). Moreover, Aβ secretion in human iPSC-derived neurons carrying *APOE⋆ ε4* is significantly higher than in those with *APOE⋆ ε3*, likely due to increased transcription of APP (*Wang et al., 2018*). Due to the different allele specificity that increase the risk of AD, the upregulation of the APOE gene expression in our result might be contributed by the allele specificity of the gene which may be caused by *APOE⋆ ε4* allele. Further studies are needed in order to prove the allele specificity of the gene. Down regulation of APOE gene- in the group treated with curcumin may suggest inhibitory effects of this compound against APOE induced production of Aβ through suppression of transcriptional and APP processing (Fig. 8).

Another interesting finding in conjunction with the degenerating effects of Aβ revealed that Presenilin 1 (*PSEN1*) expression was down-regulated in the group treated with piperine. In regards to the formation of Aβ, sequential cleavage of APP by β-secretase (*BACE-1*) and γ-secretase resulted in Aβ1–40 and Aβ1–42, which is widely perceived as neurotoxic (*Tan & Gleeson, 2019*). The gene *PSEN1* is presumably the catalytic core of the enzyme and is one of the components of γ-secretase (*Maia & Sousa, 2019*). Mutations in *PSEN1* gene are associated with some incidents of early-onset familial AD (*Ghani et al., 2018*). Presenilin 1 is a substrate for glycogen synthase kinase-3β (GSK-3β) that can

phosphorylate *PSEN1*, thereby modifying its activity (*Chu & Liu, 2018*). Increased GSK-3$\beta$ expression was associated with AD as well. For example, mutations in mice that encodes these genes have resulted in elevated levels of A$\beta$ deposition, as well as learning and memory impairments (*Myers & McGonigle, 2019*; *Zhao et al., 2019*). We found that for the first time, piperine does appear to affect the activity of $\gamma$-secretase, by decreasing the expression of the catalytic component of the enzyme *PSEN1* (Fig. 8). This mechanism may result from the inhibition of GSK-3$\beta$, which generally phosphorylates *PSEN1* to stimulate $\gamma$-secretase (*Hamann, 2018*). Therefore, it can be suggested that inhibition of the APP maturation process could account for the observed decrease in A$\beta$ by interrupting the pathway that leads to its production.

## Neuroprotective effects of curcumin and piperine against neurotoxicity

We discussed earlier in the previous section from our data that curcumin and piperine exerted their effects against A$\beta$ by modulating the pathway of A$\beta$. However, if the damage had already begun in the brain, the strategy was to prevent it from progressing rapidly. Pivotal roles of these compounds against the degeneration effects caused by A$\beta$ can be explained below based on the changes in the gene expression levels.

(i) *Restoration of synaptic loss via synaptic modulation*

A$\beta$ accumulation and the loss of synapses are the main notable features of AD. Numerous research demonstrates a decrease in synapse-related proteins, with one of the most robust synaptophysin or synaptophysin-like 1 (*SYN, SYPL1*) genes being downregulated (*Ozcelik et al., 1990*; *Yang, Frendo-Cumbo & MacPherson, 2019*). Synaptophysin is an essential glycoprotein membrane of 38 kDa originally derived from presynaptic vesicles (*Wiedenmann & Franke, 1985*). Our results showed the downregulation of *SYN* gene in A$\beta$ group which coincides with previous reports (*Ozcelik et al., 1990*; *Reddy et al., 2005*). A$\beta$ peptides interfere with both pre- and post-synaptic mechanisms of glutamatergic neurotransmission (*Lacor et al., 2004*) The presence of A$\beta$ peptides located in the spines of dissociated hippocampal cells originally proposed that it could influence post-synaptic functions directly. This resulted in the assumption that the impacts of A$\beta$ peptides in synaptic dysfunction might result from an agonist action of NMDARs (*Molnár et al., 2004*). This theory was further endorsed by the results that A$\beta$ was located in the brains of AD patients at the post-synaptic ends (*Koffie et al., 2009*). Increased expression of *SYPL1* gene in the cells treated with curcumin, suggested the protective role of this compound by reversing the effect of A$\beta$ on synaptic loss (Fig. 8).

(ii) *Restoration of ubiquitin proteasome system (UPS) pathway*

The ubiquitin-proteasome system (UPS) is a key mechanism of the degradation of intracellular proteins. Impairment of the UPS has been linked in the pathogenesis of a broad range of neurodegenerative disorders, such as Alzheimer, Parkinson, and Huntington (*Gong et al., 2006*). The effect of the UPS in these diseases may be associated with deficiencies in the clearance of misfolded proteins that lead to intracellular protein aggregation, cytotoxicity and cell death. Ubiquitinated proteins are identified in oligomeric A$\beta$ plaques and neurofibrillary tangles, and a mutation in the ubiquitin (Ub) mutant protein (Ubb[+1])

causes neuronal deterioration and is connected to AD and impairment of spatial memory (*Van Leeuwen, Hol & Fischer, 2006*).

In neurons, the signaling pathway of cyclic adenosine 3′, 5′-monophosphate (cAMP)-cAMP-dependent protein kinase (PKA)-cAMP response element-binding protein (*CREB*) is involved in synaptic plasticity and cognitive function and is regulated by UPS by degrading the regulatory subunit of PKA (*Vitolo et al., 2002*). PKA activation and rise in *CREB* phosphorylation are crucial for the development of stable long-term memory (*Chain, Schwartz & Hegde, 1999*; *Fioravante & Byrne, 2011*). We demonstrated from our study that the expression of *CREB* was downregulated in the group treated with Aβ. This finding was supported by a previous study where this signaling pathway has been shown to be impaired by Aβ in cultivated cells or brain slices treated with oligomeric Aβ and in vivo as demonstrated by mouse models of transgenic AD (*Vitolo et al., 2002*; *Smith et al., 2009*). Whereas, the level of *CREB* expression was higher in the group treated with piperine, suggesting the protective role of this compound against Aβ-induced impairment via UPS pathway (Fig. 8).

(iii) *Programmed cell death as a defense against neuronal insults*

Several studies have shown that apoptotic mechanisms are activated within the AD brain. Apoptosis is characterized by blebbing of plasma membranes, nuclear condensation and fragmentation of DNA (*Metzstein, Stanfield & Horvitz, 1998*) and is triggered by a family of aspartate proteases, known as caspases, which are activated by proteolysis from pro-caspases to their active form (*Thornberry & Lazebnik, 1998*). There are currently two significant apoptosis pathways: the death-receptor pathway where caspase-8 plays a crucial initiator role and the mitochondrial pathway incorporating oxidative stress and caspase-9 activation. Caspase-8 is the wide characterized initiator caspase, which was involved in the receptor cell death program of Fas/CD95 or tumor necrosis factor (TNF). In this context, caspase-8 is considered to be the most apical member of the caspase family recruited by adapter proteins (e.g., Fas associated death domain, *FADD*) and converted by autoproteolysis into an active form (*Muzio et al., 1996*). Cleavage of Caspase-8 results in two 11 and 18 kDa active fragments, both which represent the enzyme's activated form. In turn, it is believed that Caspase-8 triggers downstream caspases, mainly caspase-3, frequently referred to as the executioner caspase. We revealed from our study that the expression level of Caspase 8 associated protein 2 (*CASP8AP2*) was upregulated in the group treated with curcumin and piperine, while *FADD* level was upregulated in a combined therapy (Fig. 8). Our significant findings speculate that the upregulation of these genes was due to a programmed cell death mechanism as a defense against neuronal damage insults caused by Aβ, such as neuroinflammation, neurotoxicity, and altered neurotransmitter release (*Panza et al., 2019*). These findings indicate that Fibrillar Aβ may induce neuronal cell death correlated with AD by triggering apoptosis after death-receptor cross-linking and concomitant caspase-8 and caspase-3 activation.

(iv) *Reversal of Aβ-induced neuronal apoptosis*

In contrast to the above mentioned activated programmed cell death in response to Aβ-induced neuronal insults, we found that, one of the pro-apoptotic gene known as Bcl-2 modifying factor (*BMF*) was down regulated in the group treated with piperine

(Fig. 8). The pro-apoptotic Bcl-2 homology 3 domain only (BH3-only) proteins are core regulators of cell death in multiple physiological and pathological conditions, including AD. The modifying factor of Bcl-2 (*BMF*) is one of those BH3-only proteins involved in the regulation of apoptosis (*Akhter et al., 2018*) through the mitochondria pathway. Our significant finding on reversal effect of piperine against A$\beta$-induced neuronal insults via modulation of *BMF* was in parallel with the previous study which reported that there was upregulation of *BMF* resulted in cell death and the BMF knockdown proved that it had protected the neurons against death evoked by A$\beta$ or NGF deprivation (*Akhter et al., 2018*).

(v) *Reversal of A$\beta$-induced neurite degeneration*

We demonstrated from our study that Adenylate kinases 5 (AK5) are down-regulated following in vitro A$\beta$ exposure in groups treated with single and combination of curcumin and piperine (Fig. 8). While, this gene was noted for upregulation in the A$\beta$ control group (without any treatment).This finding coincided with the previous study reported by *Moon et al. 2017*) which showed that AK5 mediated neurite degeneration by the reactive oxygen species (ROS) (*Moon et al., 2017*). They found that the *AK5* expression level was significantly upregulated in the primary neuronal cells exposed to A$\beta$ and hydrogen peroxide (H$_2$O$_2$). The *AK5* gene plays a primary role in the metabolism of nucleotide through nucleotide phosphoryl exchange. There were limited findings on *AK5* gene and its involvement in AD, which may provide novel insights on the reversal mechanism of a potential compound of curcumin and piperine combating this degeneration effects.

## Novel putative genes involved in AD pathway provide an opportunistic approach for future study

We demonstrated in our data, regardless of emphasizing the major extrinsic A$\beta$ pathway, we identified 25 differentially expressed genes that are involved in the AD pathway. Out of these 25, six genes that were altered in all groups were *APOE, FADD, LRP1, PLCB3, PLCB4* and *GRIN2A*. Genes *APOE, FADD, LRP1* and *PLCB3* appeared to be up-regulated in all three groups while *PLCB4* and *GRIN2A* were downregulated. We observed that *PLCB3* and *GRIN2A* genes were the genes that have limited supporting works of literature in AD. Furthermore, we suggest other novel putative genes such as *ITPR1, GSK3B, PPP3CC, ERN1, APH1A, CYCS* and *CALM2* to be investigated in the future. Interestingly, we revealed in the network analysis that genes in AD pathway such as calcium signaling pathway, neuroactive ligand–receptor interaction, long-term potentiation, apoptosis, cholinergic synapse and inflammation were also involved. These network interactions and biological pathways provide an insight and opportunistic approach to further investigate the causal role in AD, and the link between different pathways.

## CONCLUSIONS

We confirmed in our data that curcumin and piperine exerted their effects against A$\beta$ by modulation of various intrinsic pathways. As demonstrated in our previous publication, we provided evidence in which the synergistic effect of curcumin and piperine was able to inhibit and reverse the detrimental effects caused by A$\beta$. In the present study, we observed

that in the groups treated with either single or combined compounds, all of them showed neuroprotective effects against A$\beta$, which supported previous literatures on curcumin and piperine in cognitive abilities studied individually.

We successfully characterized potential genes that appeared to be involved in A$\beta$ pathway and interestingly, our data provided evidence of the anti-amyloidogenic potential of curcumin and piperine against A$\beta$. In a future perspective, it is relevant to implement this data in primary hippocampal neurons, or current approach, three dimensional model, which offers better insights towards changes in A$\beta$ as progression into AD. In addition, healthy brain cells can be included as a control to identify gene expression related to neurodegeneration and dementia. The significance of our finding was that these data may help to understand the fundamentals of disease heterogeneity at a molecular level and provide a basis before experimenting in in vivo models. Furthermore, the next step from this finding is to either inhibit or mimic selected potential genes to further investigate the changes or effects from a downstream level, considering appropriate therapies based on recognition of different target phenotypes.

### Funding
This study was supported by Taylor's University flagship research grant (Project Code: TUFR/2017/002/04) and Shibaura Institute of Technology for the publication charges. The funders had no role in study design, data collection and analysis, decision to publish, or preparation of the manuscript.

### Grant Disclosures
The following grant information was disclosed by the authors:
Taylor's University, Malaysia: TUFR/2017/002/04.
Shibaura Institute of Technology, Japan.

### Competing Interests
The authors declare there are no competing interests.

### Author Contributions
- Aimi Syamima Abdul Manap performed the experiments, analyzed the data, prepared figures and/or tables, and approved the final draft.
- Priya Madhavan, Adeline Chia and Koji Fukui conceived and designed the experiments, authored or reviewed drafts of the paper, and approved the final draft.
- Shantini Vijayabalan performed the experiments, prepared figures and/or tables, and approved the final draft.

### Data Availability
Data are available at NCBI GEO: GSE143998.

## Supplemental Information

Supplemental information for this article can be found online at http://dx.doi.org/10.7717/peerj.10003#supplemental-information.

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
