# Peer review of "Explicating anti-amyloidogenic role of curcumin and piperine via amyloid beta (Aβ) explicit pathway: recovery and reversal paradigm effects"

_PeerJ, doi:10.7717/peerj.10003_

## Round 0.1 · original submission · Major Revisions

Your manuscript has now been seen by 2 reviewers. You will see from their comments below that they find your work of interest, and some constructive points are worth considering. We, therefore, invite you to revise and resubmit your manuscript, taking into account the points raised. Please highlight all changes in the manuscript text file.

Reviewer 1 ·

Basic reporting

The current study by Manap et al. identifies numerous biological pathways that are differentially regulated in Aβ amyloid-induced SH-SY5Y cells by treatment with the individual compounds Curcumin, Piperine, or a combination of both. This study builds on their previous work (Manap et al., 2019) where they showed that Curcumin and Piperine act in synergy to confer protection from Aβ toxicity. Now, the new results provide a comprehensive gene expression profile comparison via microarray analysis between Aβ amyloid-induced SH-SY5Y cells and the three treatment conditions. The results suggest that these compounds act on multiple pathways of relevance to Aβ toxicity including production, trafficking and clearance of amyloids, as well as downstream effects on neuronal health and function. The authors mine existing literature to provide potential mechanisms by which some of these gene expression changes may confer protection from disease progression. For the remaining significant differentially expressed genes are suggested as potential starting points for further exploring unknown mechanistic details of the disease.
The study has included ample references and background literature to support their conclusions. However, there are multiple areas for improvements (outlined below) that would enhance the general readability and accessibility of the manuscript.

Experimental design

The question is well defined and how the results address it are stated clearly, including identification of potential new avenues to address existing knowledge gaps. The Methods are also described in sufficient details.

Validity of the findings

Overall it is a well-designed study with appropriate methodology and analysis, including statistical significance of the results. A large part of the study relies on summarizing existing literature for theorizing on their own findings but they also provide several new potential avenues for research into Aβ toxicity, e.g. the ‘novel’ genes ITPR1, GSK3B, PPP3CC, ERN1, APH1A, CYCS and CALM2.

There are a few concerns to be addressed:
i. The Microarray analysis identifies APOE as a gene whose expression is significantly reduced by Curcumin treatment. In the Discussion (Lines 452-468), the authors discuss in detail the role of the APOE*ε4 allele in risk for AD disease and also mention “We demonstrated that APOE*ε4 gene expression was upregulated in the group treated with Aβ alone, which may increase the risk for AD” (Line 459). However, it is not clear how the authors arrived specifically at the ε4 allele in their results. The microarray results do not suggest the specificity of the allele being upregulated, and APOE allele specificity is also typically unknown in cell lines including SH-SY5Y.

ii. Line 525: “We demonstrated from our study that the expression of CREB was downregulated in the group treated with Aβ”. In this line and a few other places, it appears that the authors are describing gene expression changes due to Aβ induction alone. However, a control condition for cells without Aβ induction was not included in the study so this is confusing. Can this be clarified?

Additional comments

Major:
i. The overall writing can be improved for better clarity and conciseness. It is currently very wordy and written in a casual and inconsistent writing style which makes it a bit difficult to read and understand in several places. Some sentences are very long whereas some are extremely abrupt and confusing. Examples: Lines 326-330, 582-585, 589-590.
ii. The Discussion section is a good effort in summarizing existing knowledge about cellular pathways involved in Aβ toxicity and where the genes identified can the current study may be potentially exerting their effect. However, it is very lengthy and in multiple places it feels like summaries of other papers e.g. Line 335 -346 appears to summarize just Yu et al., 2005.
iii. Figure 12 is practically identical to Figure 3 from Xu et al., 2014. While the reference has been cited appropriately in the figure legend, it only serves to provide the background for how PICALM may be relevant. There is no reason to reproduce a published figure from another study where no new information or insight is being added. Similarly, Figure 13 summarizes the findings of Russel et al., 2012 without advancing existing knowledge.

Minor:
i. The current version has individual experiments as individual figures, more like a thesis structure. For example, Figures 2 and 3 both demonstrate the presence of Aβ in the cells and involve microscopy images. These could be easily combined into a single figure as separate panels. Similarly, Figures 7, 8 and 9 could be panels in a single figure or at least two could be combined.
ii. The labels of Figure 5 are too small to read.
iii. Lines 406-409 give the slightly misleading impression that these are hypotheses tested in the study. It should be stated more appropriately as background knowledge that is relevant to and supported by the new findings.
iv. The first two sections under results are written like Materials and Methods. Please considering describing these better and with appropriate subtitles indicative of the result being reported.
v. Errors: Line 436 – I believe ‘components’ should be replaced by ‘substrates’. The current sentence is very tricky to understand. Also, PCA is “principal component analysis”.

Reviewer 2 ·

Basic reporting

The authors elucidated gene expression profiles in a AB-42 induced neuroblastoma cell line in response to circumin and piperine treatment. Overall, it is a comprehensive study identifying putative genes that might contribute to the development of AD.

The authors were careful to reveal the shortcomings of microarray analysis with respect to false positives and negatives in their introduction. They might want to consider to reiterate the same in the Discussion.

Experimental design

Research question is well defined and putative hits analysed thoroughly. Future work in primary neurons is required to confirm the validity of these findings as well as elucidate the mechanistic details of the molecular pathway/s.

Validity of the findings

1. The authors might consider explaining why pretreatment of the cell line with the compounds was performed prior to addition of AB.
2. Micrarray analysis of control neuroblastoma cells pre treated with circumin/ piperine/circumin +piperine might be a critical control and might be considered
3. Downregulation of RAB 5B was observed in all 3 treatment groups in response to AB addition. Downreguation of RAB 5 will lead to severe defects in intracellular protein trafficking ultimately leading to cell death. Did the authors observe any cell death over time?

---

## Round 0.2 · Minor Revisions

Your manuscript has now been seen by the reviewers. You will see from the comments below that some constructive points are worth considering. We therefore invite you to revise and resubmit your manuscript, taking into account these constructive suggestions.

Reviewer 1 ·

Basic reporting

The revised version of this manuscript has been significantly modified by the authors based on the recommendations made in the original review. The overall writing has been improved and a lot of the extra details have been trimmed to make it somewhat more concise compared to the original version.
While most issues have been taken care of appropriately, I have a few specific suggestions for improvement as outlined below.

Experimental design

N/A

Validity of the findings

N/A

Additional comments

Thank you for addressing most of the original concerns suitably. There are three minor areas where this paper could still improve further to meet the publishing quality of PeerJ:

1. Abstract: It would be really helpful to have a compact and focused abstract outlining just the key findings of the study, in context; and excluding all superficial detail.

2. The PICALM figure (now Figure 9) as pointed out before, is practically identical to a figure from Xu et al., 2014. If it is considered essential to the current study, please make sure to cite the original source in your figure legend and say something like 'adapted from'.

3. The formatting of the figures could be improved. For example, all panels in Figure 2 look exceptionally stretched out horizontally. Please ensure that images are resized properly maintaining scaling in both dimensions. Also, in Figures 4, 6 and 7, the details and text in the figure are almost impossible to read. Please fix these for the final version.

Reviewer 2 ·

Basic reporting

My concerns have been addresses satisfactorily.

Experimental design

NA

Validity of the findings

NA

---

## Round 0.3 · accepted · Accept

Thank you for the revised manuscript and response letter. I am pleased to inform you that your manuscript has been accepted for publication in PeerJ.